# CNN-based Landmark Detection in Cardiac CTA Scans

**Julia M. H. Noothout**
Image Sciences Institute
University Medical Center Utrecht

**Bob D. de Vos**
Image Sciences Institute
University Medical Center Utrecht

**Jelmer M. Wolterink**
Image Sciences Institute
University Medical Center Utrecht

**Tim Leiner**
Department of Radiology
University Medical Center Utrecht

**Ivana Išgum**
Image Sciences Institute
University Medical Center Utrecht

## Abstract

Fast and accurate anatomical landmark detection can benefit many medical image analysis methods. Here, we propose a method to automatically detect anatomical landmarks in medical images.

Automatic landmark detection is performed with a patch-based fully convolutional neural network (FCNN) that combines regression and classification. For any given image patch, regression is used to predict the 3D displacement vector from the image patch to the landmark. Simultaneously, classification is used to identify patches that contain the landmark. Under the assumption that patches close to a landmark can determine the landmark location more precisely than patches farther from it, only those patches that contain the landmark according to classification are used to determine the landmark location. The landmark location is obtained by calculating the average landmark location using the computed 3D displacement vectors.

The method is evaluated using detection of six clinically relevant landmarks in coronary CT angiography (CCTA) scans : the right and left ostium, the bifurcation of the left main coronary artery (LM) into the left anterior descending and the left circumflex artery, and the origin of the right, non-coronary, and left aortic valve commissure. The proposed method achieved an average Euclidean distance error of 2.19 mm and 2.88 mm for the right and left ostium respectively, 3.78 mm for the bifurcation of the LM, and 1.82 mm, 2.10 mm and 1.89 mm for the origin of the right, non-coronary, and left aortic valve commissure respectively, demonstrating accurate performance.

The proposed combination of regression and classification can be used to accurately detect landmarks in CCTA scans.

## 1  Introduction

Fast and accurate automatic landmark detection is beneficial for many medical image analysis methods, such as those requiring specific anatomical seed points or landmark-based image alignment. Accurate automatic detection of anatomical landmarks in medical images is challenging due to

1st Conference on Medical Imaging with Deep Learning (MIDL 2018), Amsterdam, The Netherlands.

anatomical variation among patients and differences in image acquisition. In recent years, machine learning has been exploited for this task and a number of methods for automatic detection of anatomical landmarks in medical images have been proposed.

For automatic landmark detection, conventional machine learning methods have been used to perform classification of voxels [1, 2, 3] or bounding boxes containing landmarks [4]. These methods typically perform classification based on computed features describing the vicinity of the voxel of interest. Hence, they exploit local image information only. Consequently, exhaustive, time-consuming search schemes are required to obtain good results [2]. To address this, classification has been combined with methods that first determine a volume of interest to reduce computational costs [1, 3]. In addition to classification, methods exploiting machine learning have been employed for regression to predict the location of the landmark points. These methods perform regression to determine the displacement from the analyzed voxel to the landmark of interest [5, 6, 7]. Furthermore, a combination of classification and regression has been used to determine landmark location, for example in Hough forests. A Hough forest only performs regression when its input has previously been classified as positive by classification nodes of the forest [8, 9]. Donner et al. [8] found that this combination of classification and regression leads to more accurate landmark detection compared to only performing regression.

The aforementioned conventional machine learning methods require predefined handcrafted features and often feature selection. Deep learning methods, such as convolutional neural networks (CNNs), automatically extract features that are most descriptive for the task at hand. For landmark localization, Yang et al. [10] employed a CNN to detect anatomical landmarks in 3D MR images using a 2.5D approach for slice-based image classification. By intersecting the axial, coronal and sagittal image slices with the highest classification output a landmark position was predicted. Zheng et al. [11] used a 3D approach and analyzed image patches of a 3D volume to detect landmarks by voxel classification with a cascade of two multilayer perceptrons. Besides classification, neural networks have also been employed for regression [12, 13, 14]. Zhang et al. [12] employed a cascade of two CNNs to detect multiple landmarks in medical images. The first CNN was trained to regress the 3D displacement between the input patch and multiple reference landmarks. The second CNN shared the same weights and architecture as the first network but contained additional layers to further model correlations between analyzed input patches. In contrast, Payer et al. [13] employed one CNN to localize multiple landmarks simultaneously. The CNN analyzed the local appearance of a single landmark and the spatial configuration of all other landmarks to output heatmaps that indicate landmark locations. Furthermore, Aubert et al. [14] and Ghesu et al. [15] used neural networks to determine the search path in an image from a selected initial location to the landmark. Aubert et al. [14] used a network to detect landmark locations in 2D radiographs. The network was trained to regress the 2D displacement from an initial input patch, chosen with a statistical shape model, to the reference landmark. To obtain the landmark location, the position of the input patch was moved iteratively, using the predicted displacements, until convergence was reached and the landmark was detected. Ghesu et al. [15] detected landmarks by using deep reinforcement learning to obtain the optimal search path in CT scans by training a network to predict the best navigation steps from an initial starting location to the landmark.

While previous deep learning methods used classification or regression to localize landmarks, in this study, we propose a method that exploits a fully convolutional neural network (FCNN) that performs regression and classification jointly. For a given 3D image patch the network regresses the 3D displacement vector from the center of the patch to the landmark location. We assume that patches that are close to the landmark have potential to localize the landmark more accurately than those that are far from it. Hence, to determine which patches are important for the localization, the network jointly classifies whether the landmark is present in a given patch. Finally, the landmark location is obtained by calculating the average landmark location, using the computed 3D displacement vectors of the analyzed image patches. During averaging, only patches that were classified as containing the landmark are taken into account. We evaluate the method with detection of six clinically relevant cardiac landmarks in coronary CT angiography (CCTA) scans, namely the right and left coronary ostium, the bifurcation of the left main coronary artery (LM) into the left anterior descending (LAD) and the left circumflex artery (LCx), and the origin of the right, non-coronary, and left aortic valve commissure. The left and right coronary ostia are cardiac landmarks that are often used to analyze the coronary arteries, e.g. in automatic coronary artery tracking [16]. Together with the aortic valve commissures they play an important role in assessment of aortic root anatomy which is used as

guidance for transcatheter aortic valve implantation (TAVI) [17, 18, 19]. Bifurcations of the coronary arteries are important landmarks in the analysis of the coronary artery tree and identification of coronary artery segments [20].

A number of methods have previously been proposed for detection of the coronary ostia and aortic valve commissures in CT. Ionasec et al. [17] detected the ostia and aortic valve commissures in 4D cardiac CT by performing voxel classification in a small image region, which is determined based on a set of training images. To improve results, voxel classification was performed on multiple image resolutions. Similar to Ionasec et al. [17] Waechter et al. [18] and Zheng et al. [19] performed detection of the ostia [18, 19] and aortic valve commissures [19] in a small search area. To obtain the local search area, they first performed segmentation of the aorta, either with marginal space learning [19] or by employing a model based approach [18]. Subsequently, Waechter et al. [18] detected the ostia by performing pattern matching in cardiac CT images while Zheng et al. [19] detected the ostia and the aortic valve commissures in C-arm CT scans by voxel classification. Bifurcations of the coronary arteries are often detected as part of algorithms that analyze arteries, e.g. general vessel tracking algorithms [20]. Zhao et al. [20] detected bifurcations in 3D vascular images by performing classification. In contrast to previous work [18, 19], the here proposed method analyzes complete images, hence no assumptions about landmark location need to be made. Furthermore, images are analyzed at a single resolution. Therefore, repeated analyses at multiple resolutions, like in [11, 15, 17], is not necessary.

## 2   Data

In this study, 198 CCTA scans were used that were acquired in our hospital as part of clinical routine. The need for informed consent was waived by the Institutional Medical Ethical Review Board. Scans were acquired with a 256-detector row scanner (Philips Brilliance iCT, Philips Medical, Best, The Netherlands) with a tube voltage ranging from 80 to 140 kVp and a tube current ranging from 210 to 300 mAs. During acquisition, ECG-triggering was applied and intravenous contrast was administered. Scans had an in-plane resolution between 0.29 and 0.49 mm, with 0.9 mm slice thickness and 0.45 mm slice spacing.

Reference landmarks were manually annotated with sub-voxel precision. In total six clinically relevant landmarks were identified: the left and right coronary ostium, the bifurcation of the LM into the LAD and the LCx, and the origin of the right, non-coronary, and left aortic valve commissure. The observers had the opportunity to scroll through axial slices and thus accurately localize the landmarks. Scans in which the landmarks could not be annotated were excluded from the dataset.

## 3   Method

We propose a method for automatic detection of anatomical landmarks that employs an FCNN for simultaneous regression and classification. The FCNN uses regression to predict 3D displacement vectors from the center of image patches to the landmark of interest. Additionally, the network jointly predicts whether the patch contains the landmark of interest. The landmark location might be determined by averaging over all 3D displacement vectors. However, displacement vectors determined in the patches closer to the landmark may be better suited for accurate prediction than the patches farther from the landmark. Hence, in this work, only those patches that are classified by the network as containing the landmark are used to determine the final landmark location.

The network is trained with all image patches available in a set of training CCTA volumes. During training, the network outputs the log-transformed displacement in the x-, y-, and z-direction from the center of the input patch to the landmark. By taking the logarithm of the distance, we ensure that input patches that are far from the landmark have less influence on large updates of network parameters compared to those that are close to the landmark. Orientation is incorporated into the displacement label by taking the negative of the log-transformed distance when the landmark is located on the ventral, coronal or right side compared to the location of the center of the input patch.

The FCNN is patch-based. That is, it takes input images of any size and analyzes them in a patch-based manner. The network architecture consists of six convolutional layers where the first three layers are each followed by a $2 \times 2 \times 2$ voxel max pooling layer with a stride of 2 voxels (Fig. 1). The six convolutional layers are followed by two pairs of fully connected layers, implemented as

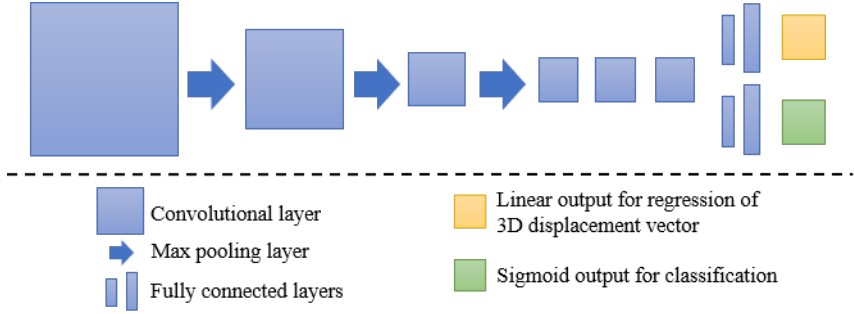

Figure 1: Architecture of the network, containing 6 convolutional layers, each with 32 ($3 \times 3 \times 3$) filters, 3 ($2 \times 2 \times 2$) max-pooling layers, 2 pairs of fully connected layers, implemented as convolutional layers with either 64 or 96 ($1 \times 1 \times 1$) filters, and finally two output layers: one for regression and one for classification. All convolutional layers employ the Exponential Linear Unit activation function. The first six convolutional layers employ zero-padding to ensure that the input and output of a layer have the same size.

$1 \times 1 \times 1$ voxel convolutions, that contain either 64 or 96 filters. All convolutional layers employ the Exponential Linear Unit (ELU) activation function [21] and furthermore, batch normalization [22] is applied. The network contains two output layers. One output layer is used for regression and one output layer is used for classification. The output layer used for regression employs a linear activation function. It has three output nodes that each predict the log-transformed distance in the x-, y-, or z-direction from the center of the analyzed patch to the landmark. The output layer used for classification employs a sigmoid activation function and has one output node that predicts whether the landmark is present in the image patch.

Due to the fully convolutional property of the network, during testing, complete images are used as input. During analysis, image patches are sampled using a grid with a spacing defined by the pooling strategy used in the network. In this study, the network contains three pooling layers with a filter size of two. Therefore, the down-sampling rate is $1/2^3$ while the patch size for this network is $2^3$ voxels. Thus, a grid with a grid spacing of eight voxels is used to sample patches from an input image.

The method was evaluated by calculating the 3D Euclidean distance between a computed landmark location and the reference landmark location.

## 4 Experiments and Results

The dataset was randomly divided into a training set containing 150 scans, a validation set containing 8 scans, and a test set containing 40 scans. The test set was not used during method development in any way. The loss function that was optimized during training consisted of two equally weighted parts: the mean absolute error between the regression output and the reference displacements, and the binary cross-entropy between the classification output and the reference labels. The network was trained for 60,000 iterations using Adam with a learning rate of 0.001 [23]. In each iteration, a mini-batch containing 25 randomly sampled subimages was provided to the network. Subimages had a size of $72 \times 72 \times 72$ voxels.

To evaluate the impact of image resolution on the analysis, scans were resized to 1 mm, 1.5 mm or 3 mm isotropic voxel size respectively. A separate network was trained for each image resolution to detect the right coronary ostium (see Appendix). Table 1 lists the results obtained by these three networks. Results are expressed as the average Euclidean distance error ($\pm$ standard deviation), as well as the range of the distance errors achieved on the test scans. The best performance was obtained with a network trained on images with an isotropic voxel size of 1.5 mm. Hence, this resolution was used in all further experiments.

To evaluate whether predictions made based on patches that are close to the target landmark are really more important for accurate localization than those based on patches that are far from the landmark, we performed an ablation study. Four networks were trained in which the log-transformation was either used or not, and the classification output layer was either used or not. When classification was

Table 1: Average Euclidean distance errors with standard deviations (Error), and the minimum (Minimum) and maximum (Maximum) distance errors expressed in mm, for the detection of the right coronary ostium. Results are obtained by networks trained on images resized to 1 mm, 1.5 mm or 3 mm isotropic voxels.

|  | Error | Minimum | Maximum |
|---|---|---|---|
| 1 mm | $2.70 \pm 2.27$ | 0.59 | 14.55 |
| 1.5 mm | $2.19 \pm 1.97$ | 0.63 | 12.72 |
| 3 mm | $3.61 \pm 2.86$ | 0.82 | 18.02 |

Table 2: Average Euclidean distance errors with standard deviations (Error), and the minimum (Minimum) and maximum (Maximum) distance errors expressed in mm, obtained by networks trained to detect the right coronary ostium in images with an isotropic voxel size of 1.5 mm. During training, the log-transformation was either used or not, and the classification output layer was either used or not.

| Log-transformed | Classification | Error | Minimum | Maximum |
|---|---|---|---|---|
|  |  | $29.07 \pm 6.83$ | 17.30 | 43.64 |
| ✓ |  | $5.57 \pm 3.35$ | 1.32 | 16.09 |
|  | ✓ | $6.33 \pm 2.54$ | 1.43 | 13.62 |
| ✓ | ✓ | $2.19 \pm 1.97$ | 0.63 | 12.72 |

omitted, we obtained the landmark location by calculating a weighted average landmark location of the predicted 3D displacement vectors, with a weight being the reciprocal of the computed 3D displacement. Again, the task was to identify the right coronary ostium. The obtained results are listed in Table 2. These results show that networks performing both classification and regression achieved better results compared to networks trained to perform only regression. Furthermore, networks trained to predict the log-transformed displacement vectors obtained better results than networks trained to directly predict displacement vectors. The best performance was achieved with the network trained to perform classification and regression of the log-transformed distance.

Finally, the proposed network was evaluated for the detection of five additional landmarks: the left coronary ostium, the bifurcation of the LM into the LAD and the LCx, and the origin of the left, non-coronary, and right aortic valve commissures (see Fig. 2). Fig 3 shows vector fields visualizing the predicted displacement vectors in three viewing planes in an image from the test set (for more results, see Appendix). Table 3 lists the Euclidean distance errors between the predicted landmark locations and the reference landmark locations. In addition, box-and-whiskers plots are shown in Fig 4. The best results were obtained for the origin of the right aortic valve commissure. Detection of the origin of the left aortic valve had the most narrow distribution. Outliers were seen during detection of the right ostium, the bifurcation of the LM, and the origin of the non-coronary, and the left aortic valve commissure.

Table 3: Average Euclidean distance errors with standard deviations (Error), and the minimum (Minimum) and maximum (Maximum) distance errors expressed in mm, obtained by networks trained to detect either the right ostium, the left ostium, the bifurcation of the LM, and the origin of the right, non-coronary, and left aortic valve commissure in images with an isotropic voxel size of 1.5 mm.

|  | Error | Minimum | Maximum |
|---|---|---|---|
| Right ostium | $2.19 \pm 1.97$ | 0.63 | 12.72 |
| Left ostium | $2.88 \pm 1.58$ | 0.18 | 7.02 |
| LM bifurcation | $3.78 \pm 2.58$ | 0.59 | 10.83 |
| Right aortic valve commissure | $1.82 \pm 0.97$ | 0.40 | 4.56 |
| Non-coronary aortic valve commissure | $2.10 \pm 0.93$ | 0.45 | 4.52 |
| Left aortic valve commissure | $1.89 \pm 0.95$ | 0.41 | 5.28 |

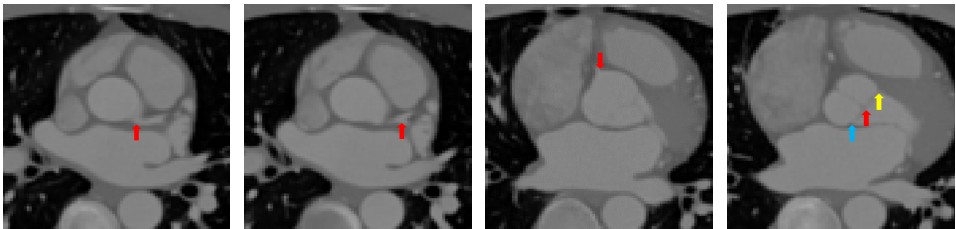

Figure 2: Axial slices from a CCTA scan, resized to an isotropic voxel size of 1.5 mm, in which reference landmark locations are indicated with a colored arrow. The landmarks shown are the left coronary ostium (left), the bifurcation of the LM in the LAD and the LCx (middle left), the right coronary ostium (middle right), and the origin of the right (yellow arrow), non-coronary (red arrow) and left (blue arrow) aortic valve commissure (right).

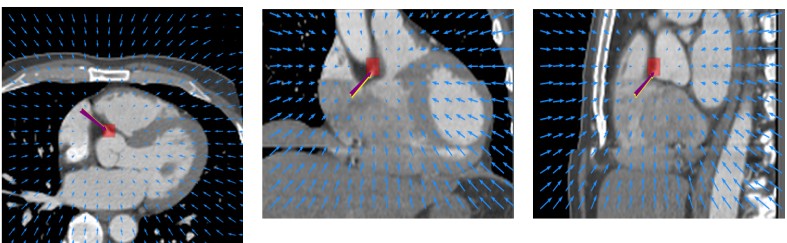

Figure 3: Vector fields visualizing the predicted displacement vectors in the axial, coronal, and sagittal plane in an image from the test set where detection of the right coronary ostium was performed. The magnitudes of the vectors should point at the right ostium, but they are rescaled for visualization purposes. The red squares indicate posterior probabilities larger than 0.5, obtained by the classification network for image patches. Reference and computed landmark annotations are indicated with a yellow and purple arrow, respectively.

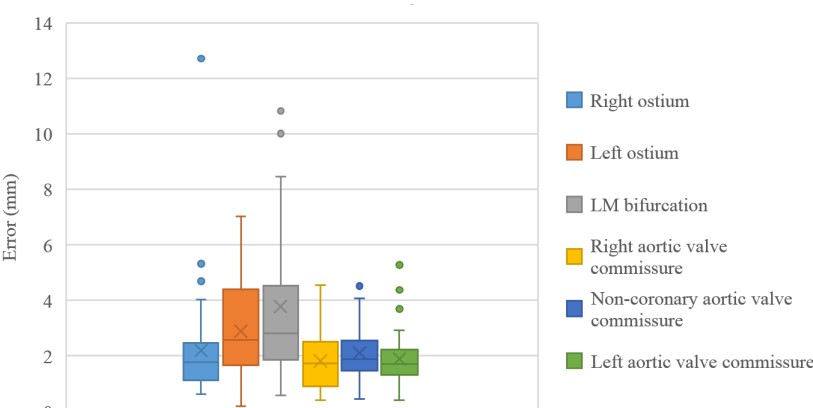

Figure 4: Box-and-whisker plot showing the errors obtained by networks trained to detect either the right ostium, the left ostium, the bifurcation of the LM, and the origin of the right, non-coronary, and left aortic valve commissure in images with an isotropic voxel size of 1.5 mm. Boxes show interquartile range. Values lower than the first quartile or higher than the third quartile are shown by lines projecting out of the box. The horizontal line within the box indicates the median value, while the average is marked by the cross inside the box. All remaining points are outliers.

# 5  Discussion and Conclusion

A method for detection of anatomical landmarks using a single FCNN that simultaneously performs regression and classification based on image patches has been proposed. Regression is used to determine the displacement vector from the center of the analyzed image patch to the landmark of interest, while classification is used to determine whether the landmark is present in the analyzed patch. Results of both tasks are combined to predict the landmark location based on patches classified as containing the landmark.

The method was evaluated for detection of six clinically relevant landmarks in CCTA scans, namely the right and left coronary ostium, the bifurcation of the LM into the LAD and the LCx, and the origin of the right, non-coronary, and left aortic valve commissure. The results demonstrate that the method is able to detect landmarks with high accuracy. The smallest average error was obtained for detection of the origin of the right aortic valve commissure, and the largest average error for detection of the bifurcation of the LM. This bifurcation is often not clearly visible as one specific point and hence, it is more difficult to accurately detect. Furthermore, outliers were seen in detection of the right coronary ostium, the LM bifurcation, and the origin of the non-coronary and left aortic valve commissures. Visual inspection of the results showed that this was often caused by anatomical deviation in the scan. For instance, the right ostium was located more dorsal or the LM bifurcation was located more ventral than in other scans in our dataset. Training the network with more scans that depict these types of variation in the anatomy could solve this. Therefore, in future work, we will increase the dataset size to ensure the presence of a large range of anatomical variations in the training and test images.

The proposed method uses simultaneous regression and classification of samples. We found that the combination of these two tasks outperformed the use of only regression. This is in line with the results of Donner et al. [8], who combined classification and regression by employing a Hough forest. Furthermore, this confirms that patches close to the landmark, i.e. those containing it, are able to localize the landmark more accurately. This also suggests that regression using patches far from the landmark is not needed to obtain the final result. Therefore, classification-based identification of patches close to the landmark and subsequent regression using only these patches may in theory lead to similar detection accuracy. Here, we chose to simultaneously perform classification and regression in the full image for computational efficiency. Moreover, our results demonstrated the importance of the log-transform on large displacements to dampen the effect of these displacements during training.

We have performed experiments evaluating landmark detection using images at three different resolutions (1 mm, 1.5 mm and 3 mm). The results showed that images resized to 1.5 mm isotropic resolution led to more accurate landmark localization than those resized to 3 mm resolution. Given the small size of the coronary arteries, images resized to larger voxel sizes likely did not allow sufficiently accurate visualization and hence, accurate localization of the landmarks. More accurate localization may be possible in images with a higher resolution. However, we found that image patches at a resolution of 1 mm might not contain sufficient context to accurately localize the landmarks. One solution could be to investigate strategies in which predictions are based on larger image patches. For this, computational limitations will need to be overcome. When successful, such strategies may also allow analysis at the native CCTA resolution, which is in the order of 0.5 mm.

Bifurcations of arteries are often detected as part of methods that analyze the arteries e.g. general vessel tracking algorithms [20]. Therefore, results are often reported as a percentage of successful detections, making comparison of our results for the detection of the bifurcation of the LM with other studies hardly possible. The obtained results for detection of the coronary ostia and the origin of the three aortic valve commissures are comparable with the results obtained in previous studies. Ionasec et al.[17] detected the coronary ostia and three aortic valve commissures in 4D cardiac CT and obtained an average error of 2.28 mm over all detected landmarks. However, these results also include results obtained for the aortic valve hinges and leaflet tips and therefore, they are not directly comparable with the results obtained in our study. Zheng et al. [19] detected the coronary ostia and three aortic valve commissures in C-arm CT scans and obtained an average error of 2.07 mm for detection of the coronary ostia and 2.17 mm for the aortic valve commissures. Waechter et al. [18] performed pattern matching to detect the ostia in cardiac CT and obtained an average error of 1.2 mm for the left and 1.0 mm for the right ostium. Taking the average of obtained distance errors, for detection of the coronary ostia and the three aortic valve commissures, we obtain an average distance error of 2.54 mm and 1.94 mm respectively. In contrast to previous work that required initial

localization of the vicinity of the target landmark [17, 18, 19] and additional segmentation of the aorta [18, 19], the here proposed method does not require any preprocessing steps. The method analyzes complete images and thus, is capable of detecting the target landmarks in large 3D image volumes with high accuracy. As demonstrated by the results, direct learning from the data without preprocessing steps incorporating knowledge about the anatomy leads to accurate detection of the six cardiac landmarks.

In this study, the method was applied for the localization of six clinically relevant landmarks in CCTA. However, the method could be readily applied for detection of other anatomical landmarks in diverse CT or MR images. Moreover, by adjusting the network architecture to analyze 2D instead of 3D images, the method would be applicable to landmark localization in 2D scans. In this work, we trained the network to detect only one landmark, but in future work it may be useful to combine detection of multiple landmarks by considering spatial relationships between different landmarks.

To conclude, the proposed method performed accurate localization of six clinically relevant anatomical landmarks in the coronary arteries and the aorta in CCTA scans. This suggests that the method may be applicable to aid in the detection of coronary artery segments or in the assessment of the aortic root anatomy as guidance for TAVI.

### Acknowledgments

This work is part of the research program Deep Learning for Medical Image Analysis with project number P15-26, which is partially financed by the Netherlands Organization for Scientific Research (NWO) and Philips Healthcare.

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

**APPENDIX**

| Original Resolution | Voxel size: 1 mm | Voxel size: 1.5 mm | Voxel size: 3 mm |
|---|---|---|---|

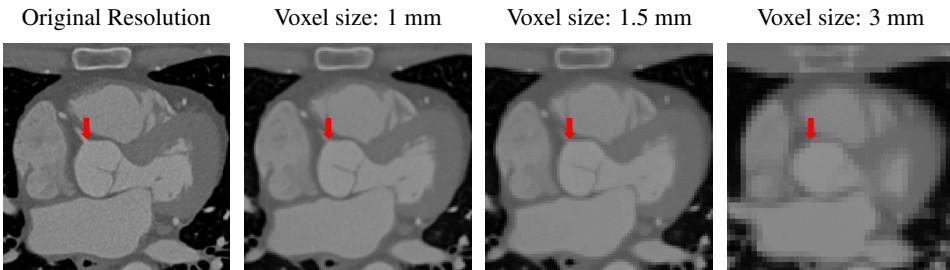

Figure 1: Axial slices from a CCTA scan, showing the right coronary ostium. The slices are shown at the original image resolution (left), resized to 1 mm (middle left), 1.5 mm (middle right) or 3 mm (right) isotropic voxels. The reference landmark location in the slices is indicated with a red arrow.

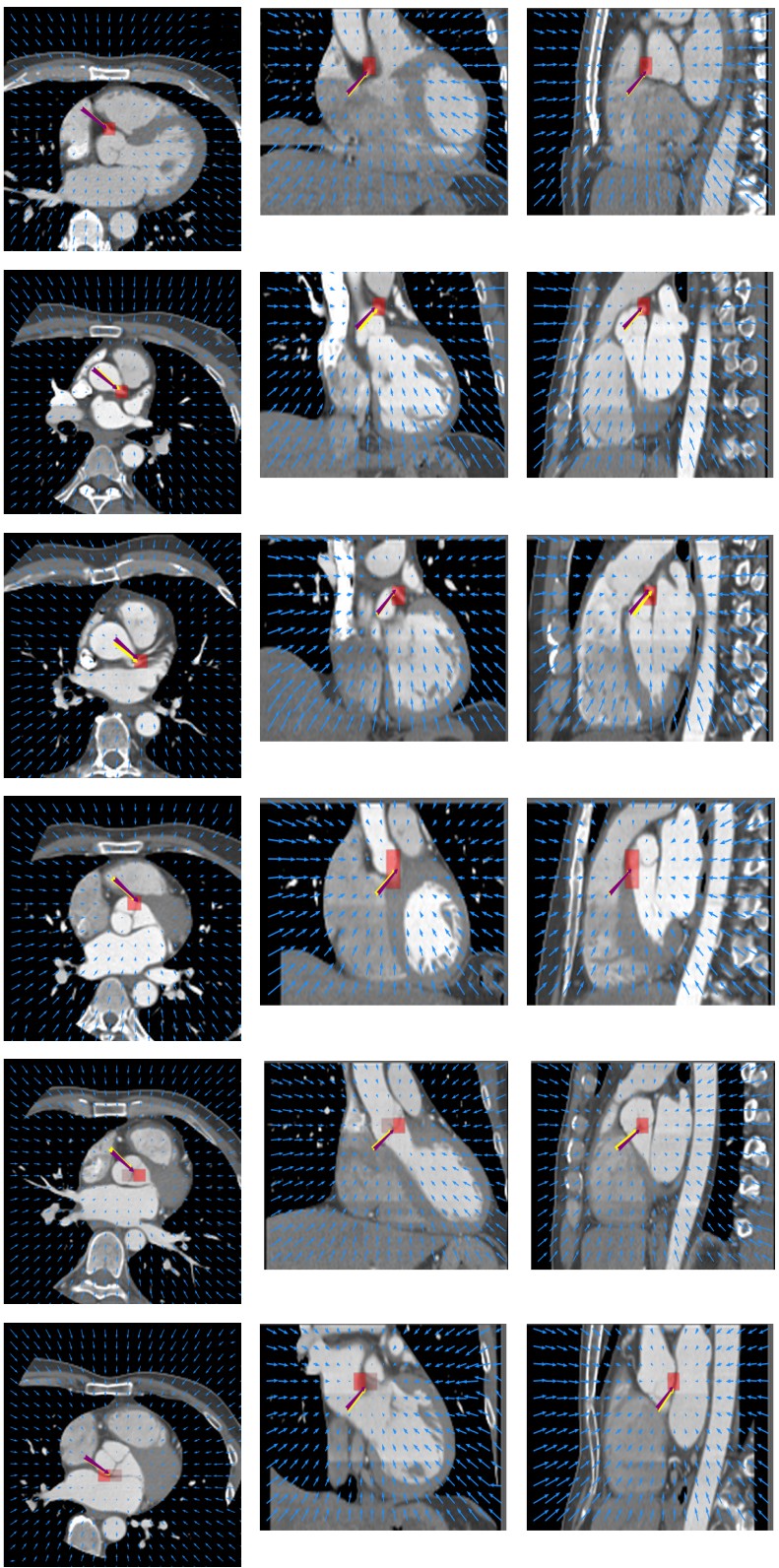

Figure 2: Vector fields visualizing the predicted localization vectors in the axial, coronal, and sagittal planes in images from the test set. The magnitudes of the vectors should point at the ostia, but they are rescaled for visualization purposes. The red squares indicate posterior probabilities that are larger than 0.5, obtained by the classification network for image patches. Reference and computed landmark annotations are indicated with a yellow and purple arrow, respectively. From the top to the bottom row, results are shown for detection of the right and left coronary ostium, the bifurcation of the LM, and the origin of the right, non-coronary, and left aortic valve commissure, respectively.