# OpenReview forum: "CNN-based Landmark Detection in Cardiac CTA Scans"
_MIDL.amsterdam/2018/Conference — MIDL 2018 Poster_

### Review · AnonReviewer2 · 2018-05-04
**The contribution of the paper is unclear**

**Rating:** 2
**Confidence:** 2

**Review:**

Pros:
+an interesting application
Cons:
-the presentation of the paper should be improved
-evaluation only on one private dataset
-unclear contribution

The paper proposes a method that for each image patch jointly produces a 3D displacement vector to the landmark and a binary classification label whether the patch contains the landmark or not.

The use of terms "patch-based" and "fully convolutional network" (FCN) is a bit unclear in the paper. FCNs are traditionally used to perform dense image prediction, where the input and output have the same dimensions (e. g. image -> segmentation, image->image, etc...).

The drawing in Fig. 1 is a bit unclear. The usage of fully connected layers suggest that there is a flattening operation in the model (going form [b, c, 0, 1, 2] feature map to [b, c*0*1*2] feature map). However, the caption and the writing suggest that there is no flattening and that fully connected layer refer to 1x1x1 convolution. Moreover, it is not clear what is a 3D displacement vector. It seems that the model outputs two tensors. if the input is a tensor of a dimension [b, c, 0, 1, 2] the outputs would two tensors of the dimensions [b, c1, 0/2^3, 1/2^3, 2/2^3] and [b, c2, 0/2^3, 1/2^3, 2/2^3]. I'd suggest modifying the figure and improving the model description.

The motivation behind the architecture choice is missing in the paper. It is not clear why the patch based approach would be the best to tackle the problem of landmark detection. Wouldn't it be better to use a dense prediction network such as FCN to predict vector displacement field for the whole volume (gong from an input of the dimension [b, c, 0, 1, 2] to an output of a dimension [b, c, 0, 1, 2])? Another alternative would be to employ object detection models (e. g. YOLO or R-CNN) to detect directly the landmark from the full volume.

The validation set is rather small. Why use only 8 scans for validation when having access to 198 scans?

The proposed method is evaluated on a private dataset. The paper could be improved by providing results on some public dataset of keypoints/landmark detection where the improvement of the proposed method could be evaluated w.r.t. alternative approaches. If no public dataset exist, the authors should justify the strengths of their approach by comparisons to other methods (e.g. object detection pipelines or FCNs).

In Table 2. How the top entry model was trained? For classification + regression model, how the mean error in mm was obtained? How false detections (classifying a position as containing a landmark when in reality there is no landmark) are handled within the proposed metric?

Since the model pools the input volume 3 times, how the results are calculated? Do the authors upsample the results to full resolution before computing the metrics?

It seems that the number of landmarks is limited in the used dataset (e. g. there are 6 clinically relevant landmarks). However, it seems that there is no way to limit the number of landmarks per volume with the proposed approach (e. g. the method can output an arbitrary number of landmarks position).  Could the authors comment on that? How we could incorporate the knowledge about number of landmarks into the model?

**Special Issue:**

No

---

### Review · AnonReviewer1 · 2018-05-07
**Joint regression and classification for localisation is a good idea but comparison with recent literature missing**

**Rating:** 3
**Confidence:** 3

**Review:**

Authors present a landmark localisation network that combines classification and regression to achieve best results. The original idea to use classification and regression for localisation was first proposed within the framework of random forests, i.e. Hough Forests. Authors here implement the same idea for neural networks and empirically show its value for the task.

Pros:
1. The article is very clearly written and the method is well explained.
2. The proposed approach is simple yet effective.
3. The authors evaluated each component of the proposed model and empirically demonstrate the advantages. For instance, they show that using log-transformation improves the results as compared to not using it. I really appreciate this!
4. Despite the fact that the idea is not very novel, I think authors did a very nice implementation for neural networks.

Cons:
1. The comparative evaluation is weak.
1a. Authors mention results from the literature on the same structures in the discussions section. However, these methods were not run on the same dataset and hence difficult to compare in terms of performance.

1b. Comparisons were mostly done with [17, 18, 19], which were published in 2008, 2010 and 2012 respectively. None of these works use neural networks and they are not recent. Surprisingly, authors cite several algorithms that use neural networks for object localisation [10-15] but do not compare their methods against these more recent contributions. As a result, it is difficult to judge the value of this contribution with respect to deep learning literature on the topic.

2. The method is patch-based but I could not find in the article the size of the patch. They mention 72^3 sub-images but it is not obvious that this is the patch size. It would be useful to provide such details for reproducibility.

3. Authors state that landmarks were manually annotated with `"sub-voxel precision". How is this possible?

Overall, I think this is a neat idea and the evaluation of the different components are convincing. However, comparative evaluation is missing, which makes it difficult to judge the value of the contribution.

**Special Issue:**

No

---

### Review · AnonReviewer3 · 2018-05-09
**Interesting patch-based landmark detection algorithm, but needs clarification**

**Rating:** 3
**Confidence:** 2

**Review:**

In this paper, the authors present a deep convolutional layer for the detection of anatomically relevant landmarks in cardiac CTA Scans. In particular, the use a multi-task approach, including regression and classification in the same patch-based architecture. Unlike other alternative approaches, they work with the entire volume, without using any additional region of interest detection previous to the landmark detection. To compensate for the potentially higher influence of those patches further from the target landmark, they use the logarithm of the Euclidean distance as metric of the loss function. Technically speaking, the paper is not particularly novel. In general, the paper is well written and properly structured, being easy to follow. The results are promising, however, there are some technical details that needs to be clarified, such as the computational cost of the proposed framework during training and testing. This is not trivial, since the proposed architecture is landmark-specific. That is, a separate network needs to be trained for each different landmark.

Strengths:
-	Interesting patch-based landmark detection method. The combination of distance regression + classification is interesting.
-	The proposed method doesn’t require a pre-define size of the input volume, or the previous delimitation of a region of interest.

Weaknesses:
-	The proposed architecture is landmark-specific. A separate network needs to be trained for each landmark.
-	It is not entirely clear to me how they include the orientation into the displacement vector.

Additional comments:

	“We assume that patches that are close to the landmark have potential to localize the landmark more accurately than those that are far from it. […] only patches that are classified as containing the landmark are taken into account.”  Is there any actual correlation between the error in the estimation of the displacement vector and the probability of containing the landmark within the patch? I guess the initial hypothesis could be easily tested. Did the authors consider to use a weighted average, using the classifier probability as weights?


	“Scans in which the landmarks could not be annotated were excluded from the dataset.”  What is the final number of scans used? In the experimental section it is said that the total number of initial scans were used (including training, validation, and test).

	“Orientation is incorporated into the displacement label by taking the negative of the log-transformed distance”  I guess it should be the negative of the exponential of the log-transformed distance.

	“[…] by taking the negative of the log-transformed distance when the landmark is located on the ventral, coronal or right side compared to the location of the center of the input patch.”  Doesn’t it require to know in advance the location of the landmark. How is this solved at testing time? I guess the images were previously aligned, then, weren’t they?

	“[…] that contain either 64 or 96 filters.”  This is a bit confusing. Please, clarify the size of the fully connected layers.

	“All convolutional layers employ the Exponential Linear Unit”  What is the advantage of ELU over the most commonly used RELU?

	Was data augmentation used during training?

	“[…] networks performing both classification and regression achieved better results compared to networks trained to perform only regression.”  Actually, according to the results shown in Table 2 for the right coronary ostium, this is only true when not using the log-transformed loss. If a logarithmic loss is used, the regression network outperform the multi-task using Euclidean distances. It I seems that it is actually the use of the logarithmic distance that makes the difference.

	Please, comment on the accuracy of the classifier included in the proposed architecture.

	“Visual inspection of the results showed that this was often caused by anatomical deviation in the scan.”  The use of data augmentation could help to improve the robustness of the method.

	The experiments to identify the optimal configuration (e.g. isotropic voxel size), were done only for the right ostium. However, are the left ostium and the bifurcation the ones who showed poorer accuracy. I would recommend to repeat the experiment for the six landmarks.

	The patch size could also be an important element in order to improve the overall landmark detection accuracy, particularly of the most challenging ones.

	Please, provide details about the computational cost (training and testing), as well as the sw and hw used (GPU, library, etc).




**Special Issue:**

Yes

---

### Decision · Program_Chairs · 2018-05-15
**Paper88 Acceptance Decision**

Poster